# Deliberate Self-Poisoning with Plants in Southeastern France, a Poison Center 20-Year Report

**DOI:** 10.3390/toxins15120671

**Published:** 2023-11-24

**Authors:** Romain Torrents, Julien Reynoard, Mathieu Glaizal, Corinne Schmitt, Katharina Von Fabeck, Audrey Boulamery, Luc de Haro, Nicolas Simon

**Affiliations:** 1Aix Marseille Univ, APHM, INSERM, IRD, SESSTIM, Hôpital Sainte Marguerite, Clinical Pharmacology and Poison Control Centre, 13009 Marseille, France; nicolas.simon@ap-hm.fr; 2APHM, Hôpital Sainte Marguerite, Clinical Pharmacology and Poison Control Centre, 13009 Marseille, France; julien.reynoard@ap-hm.fr (J.R.); mathieu.glaizal@ap-hm.fr (M.G.); katharina.moenikes@ap-hm.fr (K.V.F.); audrey.boulamery@ap-hm.fr (A.B.); luc.deharo@ap-hm.fr (L.d.H.)

**Keywords:** plants poisonings, suicide attempts, antidote

## Abstract

Introduction: In a few regions of the globe, deliberate botanical intoxication may induce significant rates of toxicity and fatality. The objective of this report was to describe plant self-intoxication using the experiences of the southeastern France poison control center (PCC) between 2002 and 2021. Results: During those 20 years, 262 deliberate plants poisonings were reported involving 35 various plants. In most of the cases, poisoning was caused by *Nerium oleander* (n = 186, 71%), followed by the *Datura* genus (4.2%), *Ricinus communis* (3.8%), *Taxus baccata* (1.9%), *Digitalis purpurea* (1.2%), *Aconitum nape* (1.9%), *Myristica fragans* (1.5%), and *Pyracantha coccine* (1.2%). Through the 262 plants poisonings, 19 patients among the 186 *Nerium oleander* poisonings received Digifab as an antidote and 1 patient received physostigmine among the 11 Datura poisonings. Only four deaths were reported for this review, each involving *Nerium oleander*. Discussion: The first involved species was *Nerium oleander* (71% of all plants poisonings), then *Datura* sp and *Ricinus communis*. It is explained by this native local species’ important repartition. Most patients must be admitted to an emergency department for adapted medical care; however, only 41 of them described severe poisonings symptoms. Even fewer needed an antidote, only 20 patients. There is no protocol for the use of a specific treatment, and it might be interesting to develop one for this purpose. Material and Methods: This retrospective review was realized with files managed by the southeastern France PCC based in Marseille from 2002 to 2021. Our department covers the complete French Mediterranean coast, Corsica, and tropical islands (Reunion Island, Mayotte). For every patient, toxicity was evaluated using the Poison Severity Score (PSS).

## 1. Introduction

Human ingestion and cutaneous or ophthalmic exposure with toxic plants are common [1]. Generally, those exposures do not involve severe intoxication [1], but in a few regions of the globe, deliberate botanical poisonings may induce significant rates of toxicity and fatality [1]. Most of the botanical intoxication can be treated symptomatically, and an antidote is usually only administered in some rare cases.

Reviews of phone calls made to poison control centers demonstrate that a substantial number of calls are caused by the consumption of plants (10% of calls) [1,2]. Fortunately, severe intoxication is uncommon [3,4]. Deaths are even more rare in the industrialized world. However, botanical intoxication is a significant medical issue in some areas of the developing world. *Thevetia peruviana* (yellow oleander) [5,6], *Datura stramonium* [7], *Cerbera manghas* (sea mango or pink-eyed cerbera [8]), and *Cleistanthus collinis* (teak species) [9,10] induce a high mortality rate every year in Asia [5,11,12].

Three possible circumstances lead to botanical intoxication: unintentional, deliberate with substance misuse, or intentional self-harming [1]. Unintentional poisoning with plants is common in young children who want to discover their surroundings by ingesting many things. Many accidental plant expositions [13,14] and many misuses [15,16] have been described, while reports on intentional intoxication are very rare [17,18]. Many are European case reports or small case series [1]. The epidemiology of human exposure to plants has only been studied in a few studies [19,20,21,22,23,24,25,26]. According to these studies, accidental exposure is the most common cause of intoxication, followed by misuse.

As a result, we realized a retrospective study to evaluate the epidemiology of suicide attempts with plants in southern France and to determine which species were principally involved in severe and lethal intoxication. The review time was from January 2002 to December 2021. The report objective was to evaluate the clinical severity of plant poisonings for human beings in southeastern France and to determine which one can really induce severe toxicity.

## 2. Results

As shown in Table 1, 262 cases of plant poisoning were collected. In most of the cases, self-poisoning was caused by *Nerium oleander*, which was responsible for 186 self-poisonings (71%). The second most used plant was the *Datura* genus (4.2%) with 11 poisonings, and the third was *Ricinus communis* (3.8%) with 10 poisonings. The other poisonings were with *Taxus baccata* (1.9%), *Digitalis purpurea* (1.2%), *Aconitum nappelus* (1.9%), *Myristica fragans* (1.5%), and *Pyracantha coccine* (1.2%). There were 35 other patients in this series, but with only one or two identical species involved: for example, the *Euphorbia* genus, the *Pittosporum* genus, *Viscum album*, and *Ilex aquifolium*. In total, 37 species caused poisonings in this report. These other plants represented 13.3% of the entire series. According to the PSS, 41 poisonings were severe, 34 of which were due to *Nerium oleander*. Only four deaths were reported, all due to *Nerium oleander*. Among all the 186 *Nerium oleander* poisonings, only 19 were given Digifab as an antidote. The only other antidote used in this series was physostigmine for one patient among the 11 Datura poisonings. All of those 20 uses of an antidote have allowed a medical recovery without sequalae.

Table 2 shows that the medium patient age was 38 +/−19.1 years. Mainly women used plants to commit suicide (55%). Few had a somatic medical history (17%), but more had a psychiatric medical history (48%). Concerning co-ingestion, many ingested benzodiazepines concomitantly (12%), and less took antidepressants (3.4%) and paracetamol (3.1%). Nearly 1/3 (N = 66, 35%) of the patients ingested the plant as a liquid (infusion or decoction). Concerning the most common symptoms, 120 patients reported nausea or vomiting (46%), 32 reported diarrhea (12%), 42 reported abdominal pain, and 27 reported drowsiness (10%). Also, 51 patients (19%) had bradycardia, 27 (10%) of whom required atropine. Most patients were hospitalized (N = 220, 85%), some of them in an intensive care unit (14%). The PSS showed severe poisoning for 16% of the patients. A Comparison of the 186 *Nerium oleander* poisonings with the 76 other plant poisonings showed significant differences. Patients who ingested *Nerium oleander* co-ingested less antidepressants (*p* = 0.02) and took them in liquid form (infusion or decoction) more often for attempted suicide (*p* < 0.01). They were more frequently symptomatic (*p* = 0.04) with nausea/vomiting (*p* < 0.01) and bradycardia (*p* < 0.01), requiring atropine (*p* = 0.04). Moreover, *Nerium oleander* poisonings seemed more severe (*p* = 0.04) than other plant poisonings. The other parameters studied were not statistically significant.

## 3. Discussion

The most incriminated species were *Nerium oleander* (71% of the report), *Datura* sp., and *Ricinus communis*.

*Nerium oleander* is an ornamental evergreen shrub belonging to the family Apocynaceae, and it is widespread in the Mediterranean area and also in subtropical and tropical regions. This plant is the most incriminated in our series and it could be explained by this native local species’ important repartition in the regions covered by our PCC. Cases of self-poisoning with *Nerium oleander* are rarely reported in the literature, as it is more often associated with accidental poisoning (children, pets). However, some cases of suicide attempts are regularly collected by toxicologists in many areas of the globe [27].

Furthermore, this is the first time that such a large case series has been reported in Europe. In 2014, Glaizal et al. [28] reported an increase in suicidal acts involving *Nerium oleander* between 2003 and 2013, collected by southeastern France’s PCC, with one death. *Nerium oleander* contains several cardiac glycosides in every piece (roots, leaves, seeds, fruits branches, stem, branches, and flowers) [29]. Oleandrin is the main important toxin of this species. It is found throughout this plant, but its concentration varies according to the age of plant and the season [30]. Glycosides are not inactivated by warming or dehydrating *Nerium oleander* leaves.

Gastrointestinal, neurological, and cardiovascular symptoms are the most common clinical signs, as shown in our case series. Episodes of nausea, epigastralgia, and emesis are usual at the onset of acute Nerium *oleander* poisonings with a physiological response to decrease the absorption of the toxins [31]. The cardiotoxic glycosides’ effects begin as heart rhythm disorders, including ventricular arrhythmia and sinus bradycardia, and can lead to death. The mechanisms responsible for the toxicity of *Nerium oleander* are like those of digitalis glycosides. It particularly inhibits the cardiomyocyte Na^+^/K^+^ ATPase, inducing a positive inotropic effect due to a hydro-electrolytic imbalance corresponding to an increase in intracellular Na^+^ and Ca^2+^ and extracellular K^+^ [31].

Few patients in our study described severe symptoms and even fewer needed an antidote. There is no protocol for the use of a specific treatment, and it might be interesting to develop one for each type of plant. The therapy for oleander poisoning is also like that of digitalis. Acute poisoning medical management involves hydro-electrolytic troubles adjustment (kalemia), transitory electrostimulation to treat a complete atrioventricular block, and the Fab antidote injection [32]. In vitro research reported that antidigoxin antibodies were able to bind and decrease oleandrin dilution [33]. Other studies have reported a connection between clinical improvement and the quantity of antidigoxin antidote injected in the human management of *Nerium oleander* poisonings.

The large dose of digoxin Fab antibodies that is required to produce a positive clinical effect is likely due to the lower affinity of digoxin Fab to binding natural glycosides. [34]. However, this antidote is very expensive, and availability can be very difficult.

In this study, every patient who received Digifab had a recovery without sequalae. This is coherent with data from the literature [1,27,33,34,35]. However, none of our patients received repeated administrations of digoxin-specific antibody fragments, as has been previously described [35].

Almost all patients were hospitalized (because of self-poisoning), and four deaths were reported for this review, each involving *Nerium oleander.*

Concerning ricin poisonings, Worbs et al. reported that the mortality was low, according to actual medical management, except for self-poisonings where a ricin-containing extract was administered, showing the more important severity of ricin with injection in comparison to ingestion [36].

Self-poisoning with *Datura* sp. is rare and not well documented. Its toxicity is related to the three alkaloids contained in the plant (atropine, scopolamine, and hyoscyamine). Usually, consumption is either accidental (cooked plant) or accidental in children, but it has been known to be deliberately taken for its hallucinogenic effects. The fatal cases published with *Datura* concerned voluntary consumption for its hallucinogenic effects. Furthermore, a study from Kerchner in 2019 [37] involving *Brugmansia* (another Solanaceae) and *Datura* intoxications (167 patients) showed that misuse poisonings (60%) were twice as high as unintentional ingestion (30%), but 10% were induced by self-poisonings.

## 4. Conclusions

The most incriminated species in this study were *Nerium oleander* (71% of this report), *Datura* sp., and *Ricinus communis*. A total of 4 patients out of 242 died in our study, and all of the deaths involved *Nerium oleander*. It is explained by this native local species’ important repartition in the regions covered by our PCC. Because of self-poisoning, 85% of the patients were hospitalized, but only 16% described severe symptoms, and 7.6% needed an antidote. There is no protocol for the use of a specific treatment in plant poisoning, and it might be interesting to develop one, especially in *Nerium oleander* poisoning.

This study also allows for a more detailed comparison with reports from other locations of the globe, for every plant reported, but especially with the oleander genus. Indeed, in some Asian countries, oleander poisoning is the most common toxin in suicide attempts with a high success rate.

## 5. Materials and Methods

Data source: Data were compiled by the Marseille PCC, which investigates all exposure reported to the southeastern France poison center. Our department covers the entire French Mediterranean coast, Corsica, and tropical islands (Reunion Island and Mayotte).

File selection: All files reporting botanical exposition and self-poisonings, from 1 January 2002 to 31 December 31 2021. Cases involving animals or unintentional botanical intoxication in children and duplicate cases with no real plant exposure (information or prevention calls) were not included in this study.

Data collected: For each case, the following parameters were described: age, sex, circumstance/location of exposure and medical background, plants involved, co-ingestion, symptoms, treatment, and evolution. The severity of intoxication was calculated using the Poisoning Severity Score (PSS) [26].

A blinded review of each file was realized twice by two medical toxicologists.

In case of divergence between the two reviews, a third review was performed by a third clinical toxicologist.

Statistics: Analysis was performed using IBM SPSS version 20 software.

Categorical data were described as percentages and quantitative variables as averages with standard deviation. The Chi-square test or Fisher exact test were used to compare distribution of qualitative variables, and the Student t-test was used to compare continuous variables. A significance threshold of 0.05 was adopted for all the statistical analysis.

## Figures and Tables

**Table 1 toxins-15-00671-t001:** Description of the cases of plant suicide attempts collected by the southeastern France PCC from January 2002 to December 2021.

Plants Involved	N (%)	Symptomatic Cases	Severe Poisoning (PSS = 3)	Death (PSS = 4)	Antidote Use
*Nerium oleander*	186 (71%)	137	34	4	19
*Datura* genus	11 (4.2%)	10	3	0	1
*Ricinus communis*	10 (3.8%)	5	1	0	0
*Taxus baccata*	5 (1.9%)	1	0	0	0
*Aconitum napellus*	5 (1.9%)	4	1	0	0
*Myristica fragrans*	4 (1.5%)	4	0	0	0
*Digitalis purpurea*	3 (1.2%)	2	0	0	0
*Pyracantha coccinea*	3 (1.2%)	0	0	0	0
Other plants (max 2 cases each)	35 (13.3%)	19	2	0	0
Total	262	182	41	4	20

**Table 2 toxins-15-00671-t002:** Comparison between self-poisoning with *Nerium oleander* and other plants collected by the southeastern France PCC from January 2002 to December 2021.

	Plant Poisonings (N = 262)	*Nerium oleander* Poisonings (N = 186)	Other Plant Poisonings (N = 76)	*p*
Medium age	38+/−19.1	38.5+/−19.2	37.2+/−18.8	0.7
Female gender	143 (55%)	104 (56%)	39 (52%)	0.3
Somatic medical history	45 (17%)	32 (17%)	13 (17%)	0.9
Psychiatric medical history	125 (48%)	89 (48%)	36 (47%)	0.9
Benzodiazepine co-ingestion	31 (12%)	21 (11%)	10 (13%)	0.7
Antidepressant co-ingestion	9 (3.4%)	3 (1.6%)	6 (8%)	0.02
Paracetamol co-ingestion	8 (3.1%)	5 (2.7%)	3 (4%)	0.4
Liquid ingestion mode	71 (27%)	66 (35%)	5 (7%)	<0.01
Symptomatic patients	182 (71%)	137 (74%)	45 (62%)	0.04
Nausea/vomiting	120 (46%)	99 (53%)	21 (29%)	<0.01
Diarrhea	32 (12%)	25 (13%)	7 (10%)	0.3
Abdominal pain	42 (16%)	31 (17%)	11 (15%)	0.8
Drowsiness	27 (10%)	17 (9%)	10 (14%)	0.3
Bradycardia	51 (19%)	48 (26%)	3 (4%)	<0.01
Atropine use	27 (10%)	24 (13%)	3 (4%)	0.04
Low blood pressure	16 (6%)	14 (7%)	2 (3%)	0.2
Hospitalization	220 (85%)	162 (87%)	58 (81%)	0.2
Intensive care	35 (14%)	29 (16%)	6 (8%)	0.1
Severe poisonings (PSS = 3)	41 (16%)	34 (18%)	7 (9%)	0.04

## Data Availability

Data is unavailable due to privacy or ethical restrictions.

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
