# Peer review of "Deliberate Self-Poisoning with Plants in Southeastern France, a Poison Center 20-Year Report"

_toxins, 2023, doi:10.3390/toxins15120671_

Round 1

Reviewer 1 Report

Comments and Suggestions for Authors

1. The length of paragraphs in Introduction is not symmetrical. Please check the Introduction section again.

2. The last paragraph of introduction is the key to this research aims and the sentences loose meaning at some point. The authors are suggested to re-write this part for better clarity.

3. Future perspective after Concluding remarks is missing. Please briefly explain the future values of the research presented in this manuscript.

4. The manuscript needs to be proofread again for some minor grammatical mistakes. Please improve the English proficiency.

Comments on the Quality of English Language

The manuscript needs to be proofread again for some minor grammatical mistakes. Please improve the English proficiency.

Author Response

Thank you for reviewing this study . Here the reply :

  1. The length of paragraphs in Introduction is not symmetrical. Please check the Introduction section again.

Lenght have been corrected

  1. The last paragraph of introduction is the key to this research aims and the sentences loose meaning at some point. The authors are suggested to re-write this part for better clarity.

This paragrah has been simplified for a better understanding

  1. Future perspective after Concluding remarks is missing. Please briefly explain the future values of the research presented in this manuscript.

Perspectives has been added as advised

  1. The manuscript needs to be proofread again for some minor grammatical mistakes. Please improve the English proficiency.

The manuscript has been reviewed by a native speaker.

Reviewer 2 Report

Comments and Suggestions for Authors

This manuscript presents interesting data on suicide attempts using toxic plants. Despite its scientific relevance, the manuscript needs several corrections.

L.20-21: I did not understand the meaning of this sentence.
L.22-23: Complete the sentence: one patient among the 11 Datura poisonings.
L.27, 112, 160, 170: Do not use italic letters for “sp”.
L.27-28: I did not understand the meaning of this sentence.
L.28: “little of them”: how many?
L.29: How many patients needed antidote?
L.36-38: This data is from Germany and USA. It is more appropriate to cite the original sources of the information mentioned in the manuscript.
L.72: From the next sentence, I guess that there were two toxicologists.
L.75-77: Describe here the statistical analysis used in Table 2, including the adopted level of significance.
L.85: Is the reference to the genus because it is an unidentified species or because there are several species of the same genus? If you grouped species, it would be interesting to list all species (even as a supplementary table) so that you can have them as a source of future reference.
L.86: What is the total number of species that caused poisoning?
L.88-90: What was the efficacy of Digifab for treating Nerium poisoning? The same for physostigmine in the patient poisoned by Datura.
L.90: Complete the sentence: one patient among the 11 Datura poisonings.
L.93, 97, 102: series? Do you mean patients?
L.97: “Nearly 1/3”: include the exact amount
L.102-108 and Table 2: I didn't understand the logic of comparing poisoning by one species of plant with the set of poisonings by other species.
L.112, 118, 123, 146, 169: oleander (lowercase letters)
L.122: “[28 10.1016/j.toxac.2014.09.034]. Already”: [28] already. This reported increase is comparing which periods?
L.127: “type of tree”: delete - this information has been confirmed as erroneous
L.139-140: A protocol for each type of plant, as poisoning is different.
L.142-146: It remains to discuss here the effectiveness of Digifab in patients in the present study and compare it with data from the literature.
L.147-148: This sentence repeats the statement in lines 139-140.
L.148-149: “high quantity of antidote necessary to have a medical improvement”???
L.151-152: “availability can be difficult or may not be available or must be sent from afar”: all the three comments are the same problem, limited availability.
L.156: Why it is exceptional? The cited reference refers just to ricin poisoning, but not poisonings by other plants.
L.160-167: It remains to discuss here the effectiveness of Digifab in patients in the present study and compare it with data from the literature.
L.165: solanaceae: Solanaceae
L.170: four patients from 186
L.172-174: Replace the adjectives by the percentages of patients.
L.174-176: This statement is vague.

Comments on the Quality of English Language

The manuscript requires English revision.

Author Response

Thank you for reviewing this study. Here the reply :

This manuscript presents interesting data on suicide attempts using toxic plants. Despite its scientific relevance, the manuscript needs several corrections.

L.20-21: I did not understand the meaning of this sentence.

Sentence removed for better understanding
L.22-23: Complete the sentence: one patient among the 11 Datura poisonings.

Done
L.27, 112, 160, 170: Do not use italic letters for “sp”.

Done
L.27-28: I did not understand the meaning of this sentence.

Modified for better understanding

L.28: “little of them”: how many?

Precised as advised
L.29: How many patients needed antidote?

Precised as advised
L.36-38: This data is from Germany and USA. It is more appropriate to cite the original sources of the information mentioned in the manuscript.

Modified as advised
L.72: From the next sentence, I guess that there were two toxicologists.

Corrected
L.75-77: Describe here the statistical analysis used in Table 2, including the adopted level of significance.

Precised as advised
L.85: Is the reference to the genus because it is an unidentified species or because there are several species of the same genus? If you grouped species, it would be interesting to list all species (even as a supplementary table) so that you can have them as a source of future reference.

Unfortunatly in most cases species is unprecised
L.86: What is the total number of species that caused poisoning?

Precised as advised
L.88-90: What was the efficacy of Digifab for treating Nerium poisoning? The same for physostigmine in the patient poisoned by Datura.

Precised as advised
L.90: Complete the sentence: one patient among the 11 Datura poisonings.

Corrected as advised
L.93, 97, 102: series? Do you mean patients?

Corrected as advised
L.97: “Nearly 1/3”: include the exact amount

Precised as advised
L.102-108 and Table 2: I didn't understand the logic of comparing poisoning by one species of plant with the set of poisonings by other species.

It had been decided because the oleander species was largely the most representedand we wanted to know if specific characteristics can be highlight
L.112, 118, 123, 146, 169: oleander (lowercase letters)

Corrected
L.122: “[28 10.1016/j.toxac.2014.09.034]. Already”: [28] already. This reported increase is comparing which periods?

I completed, 2003 and 2013
L.127: “type of tree”: delete - this information has been confirmed as erroneous

Corrected
L.139-140: A protocol for each type of plant, as poisoning is different.

Precised as advised
L.142-146: It remains to discuss here the effectiveness of Digifab in patients in the present study and compare it with data from the literature.

Precised as advised
L.147-148: This sentence repeats the statement in lines 139-140.

Removed
L.148-149: “high quantity of antidote necessary to have a medical improvement”???

Changed to :The large dose of digoxin Fab antibodies that was required to produce a positive clinical effect is likely due tothe lower affinity of digoxin Fab to bind natural glycosides.[34]

L.151-152: “availability can be difficult or may not be available or must be sent from afar”: all the three comments are the same problem, limited availability.

Corrected

L.156: Why it is exceptional? The cited reference refers just to ricin poisoning, but not poisonings by other plants.

Corrected

L.160-167: It remains to discuss here the effectiveness of Digifab in patients in the present study and compare it with data from the literature.

Done
L.165: solanaceae: Solanaceae

Corrected
L.170: four patients from 186

Corrected
L.172-174: Replace the adjectives by the percentages of patients

Done

L.174-176: This statement is vague.

Modified for a better understanding

Reviewer 3 Report

Comments and Suggestions for Authors

Dear Editors,

Thank you for the opportunity to review "Deliberate self-poisoning with plants in the Southwestern France, a poison centre 20-year report".  This is a decent paper with some insignificant language related things that will easily be corrected by the technical editors. The conclusions are good and will be interesting for "Toxins" readers. I really have nothing to contribute other than it would have been nice to have seen a more detailed comparisoin of this work with reports from other locations. I have read that oleander seed is the most common toxin in suicide attemenpts in some Asian countries, though it has a low sucess rate. This paper will be a good contribution to Toxins. Thank you again for this opportunity.

Author Response

Thank you for reviewing this study. 

Round 2

Reviewer 2 Report

Comments and Suggestions for Authors

The authors have corrected the manuscript accordingly. I have no other considerations.